# CAUSAL POLICY RANKING

## ABSTRACT

Policies trained via reinforcement learning (RL) are often very complex even for simple tasks. In an episode with $n$ time steps, a policy will make $n$ decisions on actions to take, many of which may appear non-intuitive to the observer. Moreover, it is not clear which of these decisions directly contribute towards achieving the reward and how significant is their contribution. Given a trained policy, we propose a black-box method based on counterfactual reasoning that estimates the causal effect that these decisions have on reward attainment and ranks the decisions according to this estimate. In this preliminary work, we compare our measure against an alternative, non-causal, ranking procedure, highlight the benefits of causality-based policy ranking, and discuss potential future work integrating causal algorithms into the interpretation of RL agent policies.

## 1 INTRODUCTION

Reinforcement learning is a powerful method for training policies that complete tasks in complex environments via sequential action selection (Sutton & Barto, 2018). The policies produced are optimised to maximise the expected cumulative reward provided by the environment. While reward maximization is clearly an important goal, this single measure may not reflect other objectives that an engineer or scientist may desire in training RL agents. Focusing solely on performance risks overlooking the demand for models that are easier to analyse, predict and interpret (Lewis et al., 2020). Our hypothesis is that many trained policies are *needlessly complex*, i.e., that there exist alternative policies that perform just as well or nearly as well but that are significantly simpler.

The starting point for our definition of "simplicity" is the assumption that there exists a way to make a "simple choice" based on randomly selecting an action from the available repertoire in an environment. We argue that this may be the case for many environments in which RL is applied. That is, an agent may be able to simplify its policy by randomizing its action selection in some states without a drastic drop in expected reward obtained. The tension between performance and simplicity is central to the field of explainable AI (XAI), and machine learning as a whole (Gunning & Aha, 2019). The key contribution of this paper is a novel causality-based method for simplifying policies while minimizing the compromise with respect to performance, hence addressing one of the main hurdles for wide adoption of RL: the high complexity of trained policies.

We introduce an algorithm for *ranking the importance of the decisions* that a policy makes, by scoring the actions it selects. The rank reflects the impact that replacing the policy's chosen action by a randomly selected action has on the reward outcome. We develop rankings based on sample-based estimates since it is intractable to compute this ranking precisely, due to the complex causal interactions between actions and their outcomes. Specifically, for each sample, a subset of policy actions are randomly selected and replaced by a random action. This process is repeated while varying the number of policy actions being replaced. In the limit that none of the policy actions are replaced, we recover the original policy.

Ranking policy decisions according to their importance was recently introduced by (Pouget et al., 2020), who use spectrum-based fault localization techniques to approximate the contribution of decisions to reward attainment. Our algorithm uses *causal effects* techniques (Pearl, 2009) to compute the ranking of policy decisions and can be applied to black-box policies, making no assumptions about the policy's training or representation. We use the same proxy measure for evaluating the quality of our ranking as (Pouget et al., 2020): we construct new, simpler policies ("pruned policies") that only use the top-ranked decisions, without retraining, and compare the reward achieved by these policies

with the original policy's one. Experiments with agents for MiniGrid (Chevalier Boisvert et al., 2018) demonstrate that pruned policies can maintain high performance and also that performance monotonically approaches that of the original policy as more highly ranked decisions are included from the original policy. As pruned policies are much easier to understand than the original policies, we consider this a potentially useful method in the context of explainable RL. As pruning a given policy does not require re-training, the procedure is relatively lightweight. Furthermore, the ranking of states by itself provides an important insight into the importance of particular decisions for the performance of the policy overall.

Our method demonstrates that causal counterfactual reasoning (Hume, 1739; Halpern & Pearl, 2005a;b) is applicable to model-free RL and opens the door for other causality-inspired methods that can further improve the interpretability and explainability of RL policies, as well as simplify them. For example, previous work has sought to identify simplified policies via regularization. Specifically, a penalty term measuring the KL-divergence between a trained policy and a random baseline is traded off against the expected cumulative reward objective (Tishby & Polani, 2010; Todorov, 2009). The main difference with our work is that we describe an offline technique that is based on causal reasoning. We envision an avenue for a potential integration of our approach with the policy regularization approach whereby causal reasoning may be leveraged in identifying a regularized policy.

## 2 BACKGROUND AND DEFINITIONS

### 2.1 REINFORCEMENT LEARNING

We use a standard reinforcement learning (RL) setup and assume that the reader is familiar with the basic concepts. An *environment* in RL is defined as a Markov decision process (MDP) with components $\{S, A, P, R, \gamma\}$, where $S$ is the set of states $s$, $A$ is the set of actions $a$, $P$ is the transition function, $R$ is the reward function, and $\gamma$ is the discount factor. An agent seeks to learn a policy $\pi : S \to A$ that maximises the total discounted reward. Starting from the initial state $s_0$ and given the policy $\pi$, the state-value function is the expected future discounted reward as follows:

$$V_\pi(s_0) = \mathbb{E}\left(\sum_{t=0}^{\infty} \gamma^t R(s_t, \pi(s_t), s_{t+1})\right). \tag{1}$$

A policy $\pi : S \to A$ maps states to the actions taken in these states and may be stochastic. We treat the policy as a black box, and hence make no further assumptions about $\pi$.

### 2.2 CREDIT ASSIGNMENT METRICS

#### 2.2.1 CAUSAL COUNTERFACTUAL REASONING

Causal counterfactual reasoning was introduced by Hume (1739), who was the first to identify causation with counterfactual dependence. The counterfactual interpretation of causality was extended and technically formalized in Halpern & Pearl (2005a;b). Essentially, an event $A$ is a cause of an event $B$ if $A$ happened before $B$ and in a possible world where $A$ did not happen, $B$ did not happen either. In this work, we adapt these concepts to RL, where an event $A$ is a decision of a given RL policy in a given state and $B$ is the success in achieving the reward.

Based on this conceptual framework, we evaluate the significance of a particular action according to its *causal contribution to reward maximization*. The causal contribution measured by the *causal effect* $C(s, a)$ of a state-action pair, which we define as the difference in expected cumulative reward obtained with respect to alternative (counterfactual) actions $a'$. Specifically, suppose an agent selects action $a_t$ in state $s_t$ on timestep $t$, then

$$C(s, a) = V_{\pi, a_t}(s_0) - \mathbb{E}_{a_t' \sim \pi_{\text{rand}}(\cdot|s_t)}\left[V_{\pi, a_t'}(s_0)\right], \tag{2}$$

where $V_{\pi, a_t}(s_0)$ is the value function associated with policy $\pi$ except at time $t$ when the action $a_t$ is selected, $\pi_{\text{rand}}$ denotes the random policy, and $a_t' \sim \pi_{\text{rand}}(\cdot|s_t)$ implies that action $a_t'$ is sampled from the policy $\pi_{\text{rand}}(\cdot|s_t)$ at state $s_t$.

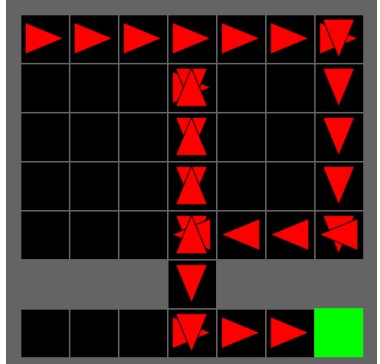 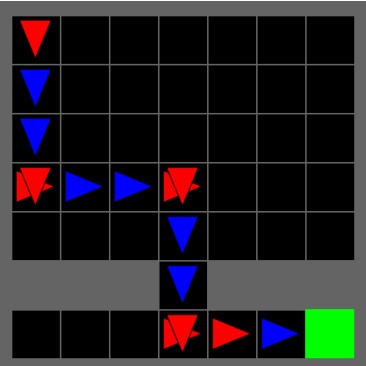

Figure 1: Traces of executions with the original policy (left) and a *pruned policy* (right). States in which we take a random action are in blue. Both policies succeed, but pruning unimportant actions simplifies the policy.

### 2.2.2 SPECTRUM-BASED FAULT LOCALIZATION (SBFL)

As a comparative ranking measure, we consider spectrum-based fault localization (SBFL). SBFL techniques (Naish et al., 2011) have been widely used as an efficient approach to aid in locating the causes of failures in sequential programs. SBFL techniques rank program elements (say program statements) based on their *suspiciousness scores*, which are computed using correlation-based measures. Intuitively, a program element is more suspicious if it appears in failed executions more frequently than in correct executions, and the exact formulas differ between the measures. Diagnosis of the faulty program can then be conducted by manually examining the ranked list of elements in descending order of their suspiciousness until the cause of the fault is found. It has been shown that SBFL techniques perform well in complex programs (Abreu et al., 2009).

The SBFL procedure first executes the program under test using a set of inputs called the *test suite*. It records the program executions together with a set of Boolean flags that indicate whether a particular program element was executed by the current test. The task of a fault localization tool is to compute a ranking of the program elements based on the values of these flags. Following the notation from (Naish et al., 2011), the suspiciousness score of each program statement $s$ is calculated from a set of parameters $\langle a_{ep}^s, a_{ef}^s, a_{np}^s, a_{nf}^s \rangle$ that give the number of times the statement $s$ is executed ($e$) or not executed ($n$) on passing ($p$) and on failing ($f$) tests. For instance, $a_{ep}^s$ is the number of tests that passed and executed $s$. There is a number of SBFL measures, based on different formulae that use these scores; some of the most popular include (Ochiai, 1957; Gonzalez-Sanchez, 2007; Jones & Harrold, 2005; Wong et al., 2007).

Recent work on the application of SBFL to RL demonstrated that SBFL techniques perform well on ranking policy decisions according to their suspiciousness score (Pouget et al., 2020). As a proxy for the quality of the ranking, the SBFL ranking was used to construct simpler *pruned* policies by only taking the high-ranked actions from the original policy and substituting low-ranked actions with a randomly generated action (see Figure 1 for a demonstration). Here, we consider a similar approach with causal counterfactual ranking.

### 2.3 RANKING POLICY DECISIONS

For SBFL ranking, we first create the test suite of mutant executions $\mathcal{T}(\pi)$ as described above. We call the set of all abstract states encountered when generating the test suite $S_\mathcal{T} \subseteq \hat{S}$; these are the states to which we assign scores. Any unvisited state is given the lowest possible score by default.

Similarly to SBFL for bug localisation, for each state $s \in S_\mathcal{T}$ we calculate a vector $\langle a_{ep}^s, a_{ef}^s, a_{np}^s, a_{nf}^s \rangle$. We use this vector to track the number of times that $s$ was unmutated ($e$) or mutated ($n$) on passing ($p$) and on failing ($f$) mutant executions, and we do not update these scores based on executions in which the state was not visited. In other words, the vector keeps track of success and failure of mutant executions based on whether an execution took a random action in $s$ or not. For example, $a_{ep}^s$ is the number of passing executions that took the action $\pi(s)$ in the state $s$,

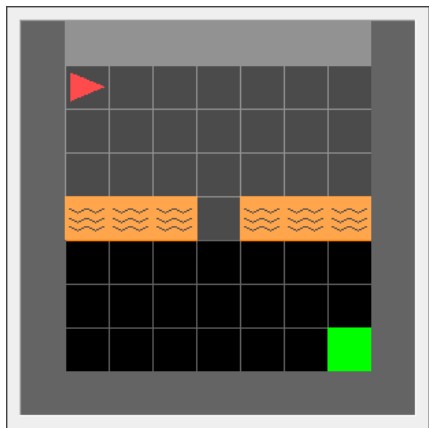

Figure 2: Minigrid lava environment.

and $a_{nf}^s$ is the number of failing executions that took a random action in the state $s$. Once we have constructed the vector $\langle a_{ep}^s, a_{ef}^s, a_{np}^s, a_{nf}^s \rangle$ for every (abstract) state, we apply the SBFL measures discussed in Sec. 2.2.2 to rank the states in $\pi$. This ranking is denoted by $rank : \hat{S} \to \{1, \dots, |\hat{S}|\}$.

We follow a similar procedure for the causal counterfactual ranking except we replace the SBFL measures with the causal effect estimate (Eqn. 2).

## 3 RESULTS

### 3.1 PERFORMANCE OF PRUNED POLICIES

The precise ranking of decisions according to their importance for the reward is intractable for all but very simple policies. We use the performance of *pruned policies* as a *proxy* for the quality of the ranking computed by our algorithm. In pruned policies, all but the top-ranked actions are replaced by randomly sampled actions. For a given $r$ (a fraction or a percentage), we denote by $rank[r]$ the subset of $r$ top-ranked states. We denote by $\pi^r$ the pruned policy obtained by *pruning* all but the top-$r$ ranked states. That is, an execution of $\pi^r$ retains actions in the $r$ fraction of the most important states from the original policy $\pi$ and replaces the rest by random actions. The states that are in $rank[r]$ are called the *original states*. We measure the performance of the pruned policies for increasing values of $r$ relative to the performance of the original policy $\pi$. To evaluate the quality of each ranking method, we measure how quickly we are able to recover the performance of the original policy $\pi$ as we reduce the set of pruned states. We start with $r = 0$, and evaluate the performance of $\pi^r$ for increasing values of $r$.

We consider how performance is recovered as the percentage of decisions drawn from the original policy increases, and as the percentage of steps in which the original policy is used increases (note that for the former, we always prune all of the states not encountered in the test suite, even at 100%). These two ways of reporting performance differ when, for example, we rank highly states that are important if visited but rare. Even if many of theses states are not pruned, the policy would still more often take a random action, despite fewer states being pruned overall.

We applied our analysis to twenty challenging minigrid environments in which the agent has to reach the goal and avoid falling into the lava (see an example in Figure 2). Our preliminary results (see Fig. 3) show that causality-based policy ranking provides a level of performance comparable, but inferior, to that of SBFL-based policy ranking in Pouget et al. (2020) (see figure captions for further details). Given that a measure of the causal effect of an action on reward attainment would seem a more direct measure of the significance of a particular action, this result highlights the surprising effectiveness of SBFL-based metrics in ranking policy decisions. However, we suggest that the limited suite of evaluations on variants of the minigrid lava environment may obscure strengths and weaknesses of each of these measures in policy ranking. For example, the large decrease in total reward (or increase in costs) received due to falling into the lava is a particularly dramatic reward

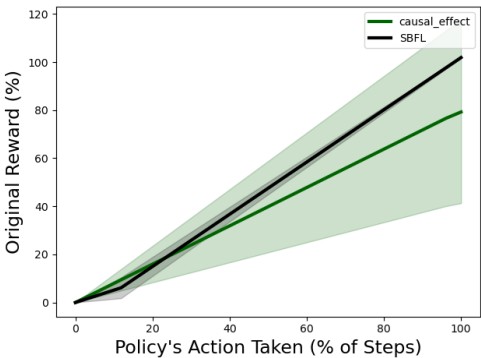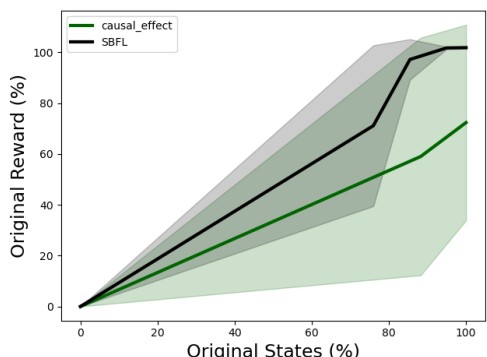

Figure 3: Comparing causal ranking and SBFL-based ranking in the minigrid lava environment based on the percentage of the original reward recovered based on a pruned policy (left). The percentage of original states occupied versus reward recovered under the pruned policy (right). Lines (shaded regions) indicate the mean (standard error) across 20 experimental runs.

contingency that any metric would be sensitive to. Thus, a fruitful avenue for future investigation, will be to design and test novel environments with subtler environment structures requiring more sensitivity from policy ranking techniques. Another concern with the current investigation is that actions are treated independently whereas it is often the case that action sequences, or even non-contiguous combinations of actions, may synergize to have a large effect on reward acquisition (Sutton et al., 1999). Extending SBFL-based and causality-based policy ranking into the multi-action domain may result in qualitatively distinct policy ranking results compared to the single-action methods.

## 4 DISCUSSION

We presented some initial results on policy simplification via ranking policy decisions based on counterfactual reasoning. The performance of the resulting policies was comparable to previously established baselines for this problem (Pouget et al., 2020). Despite the lack of an overall performance improvement, we suggest that causality-based policy ranking provides a more direct and interpretable methodology. Furthermore, we argue that this work establishes the potential for causality-based policy ranking with a rich potential for future work. Beyond causal effects measures, more sophisticated causal inference algorithms such as the PC and FCI algorithms may be deployed in the service of policy ranking (Pearl, 2009). Such methodologies may, for example, be able to adjust for complex interactions between decisions across timepoints. This could account for higher-order dependencies in policy decisions whereby combinations of actions (or action avoidances) in different states may be identified as contributing significantly to reward maximization. Additionally, we hypothesize that recent developments in score-based causal analysis (Zheng et al., 2018) may allow our policy ranking method to be extended to continuous control problems.

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
