# OpenReview forum: "Causal Policy Ranking"
_ICLR.cc/2022/Workshop/OSC — ICLR2022 OSC  Poster_

### Official Review · Reviewer_mCfP · 2022-03-15
**A potentially useful idea that needs to specify more details, and could benefit from better baselines**

**Rating:** 2
**Confidence:** 3

**Review:**

This paper proposes to evaluate a causal-intervention approach to understanding the effect of different decisions on a policy's reward. The approach is somewhat interesting, but more baselines would help illustrate the value of this approach, and more clarification of details is needed. I consider this paper borderline-reject unless the following details are clarified, but I am scoring it a 2 because there is no 1.5 option.

Comments:

* The authors should clarify in detail what the algorithms and environments they used are.
   - The authors say that they treat the policy as a black box and make no further assumptions about it, but it is necessary to at least report the details of what they used for the experiments in the paper, in order to understand how to interpret the results. For example, in the case that the algorithm was trained with a very low exploration temperature (such that it would be fairly overfit to a particular sequence of actions at convergence, even if other sequences are equally good) vs. very high temperature training might substantially change the conclusions of their approach, by changing how disruptive the intervention is.
  - similarly, the environments clearly affect the comparisons (e.g. the number of bottleneck states in them, the magnitude of the rewards for different outcomes, etc.) as the authors note in passing. Describe these details in the supplement!

* It seems that the authors would like to claim that there is an essentially *causal* aspect of their work—that is, the fact that it involves an intervention—that is essential to the contribution. They compare to another interventional approach based on SBFL. However, one could ask whether a non-interventional approach could achieve the same benefits. It would be ideal to compare to a non-interventional baseline, but at least some of these issues should be discussed. For example:
  - Assuming the authors are using a Q-learning approach (which is unclear, see above), the relationships between Q-values of the agent itself in a state are effectively estimates of how important it thinks actions are in that state. Could statistics calculated from the Q-values, without having to actually *do* the other actions, suffice for evaluating the importance of actions?
  - Relatedly, there is a resemblance between the counterfactual quantity the authors propose, and the advantage function used in algorithms like A2C, with the chief difference being whether the expectation in the second term is over a random policy or the agent's policy. Nevertheless, one might imagine that a similar calculation could be done with the advantage function.
  - If the authors are using a policy approach that does not compute action values, a similar ranking of importance could be done with statistic over the action probabilities.
  - These approaches would require the policy to not be a completely black box (at least at the policy outputs), and perhaps that is why the authors neglect them (although it would be possible to estimate the policy distribution empirically by repeatedly sampling in the same state). However, the motivation for strictly black-box policies seems unclear to me; the settings in which it is possible to sample from a policy but not to evaluate its action logits seems quite contrived. The authors should at least motivate this setting better in the paper. And even if this is the ultimate goal, it would still be useful to know how privileged-information comparisons which can access the action values/logits/distribution would compare.

---

### Official Review · Reviewer_W5PV · 2022-03-16

**Rating:** 2
**Confidence:** 2

**Review:**

Overall, I enjoyed reading the paper.
The idea was well-presented and could be understood.

Pros:
* Interesting idea to incorporate counter-factual reasoning into credit assignment
* Approach was useful to prune policies to produce a simple and effective policy

Cons:
* Design choices behind Equation 2 are not justified. For example, the approach could have evaluated the value of a policy pi with action at t wrto each of the remaining action at time t. This also yields a metric that measures causal contribution of action at t. It would be more informative to discuss the motivations behind deriving this equation.
* "Our hypothesis is that many trained policies are needlessly complex..." - this statement does not have any justification and without it, it is hard to agree with this statement.
* The paper needs to discuss how such a method can scale to domains with uncountably many states / domains with function approximation. Also, are there ways to combine this approach with existing approaches for credit assignment (e.g., eligibility traces)?

---

### Decision · Program_Chairs · 2022-03-24

Accept (Poster)